

# Seat Booker
## Office Space Reservation System



**Authors**: Jakub Chodyła⊙ · Julia Rott⊙ · Konrad Karoński⊙ · Krzysztof Urban⊙

**Supervisor:** Oleksandr Yeroshkin⊙

### Abstract

Our project, Seat Booker, is designed to ease the transition as companies return to the office, often adopting hybrid work models, by offering a system where workspaces can be reserved. In this flexible setting, employees no longer have permanent seats - instead, they can book workstations for specific days. This setup enhances both the employee and company experience by offering customizable features. Our office space reservation system allows organizations to configure offices with reservable seats. Employees can easily book a dedicated workspace for a specified day, ensuring they have a tailored work environment to suit their needs. Admins can customize workstations by adding elements like extra monitors. This flexibility ensures that employees can reserve spaces equipped with the tools they need to perform their best.

## 1 DEVELOPMENT

### 1.1 Introduction

As Smite et al. (2022) highlight, "the post-pandemic times are associated with empty offices, confused managers and organizational leaders not knowing what to do with the often-expensive rental contracts." [1] This observation reflects the broader challenges organizations face in managing resources and adapting to hybrid work environments. The pandemic has not only accelerated the adoption of remote work but also reshaped expectations for how office spaces are utilized. As Tahlyan et al. (2022) suggest, many of these trends are here to stay, making effective workspace management systems essential for modern organizations. [2]

The shift towards hybrid work models has fundamentally transformed the way companies utilize office spaces. Traditionally, employees occupied fixed seats, but the flexibility of hybrid models means many now share workspaces or visit the office only part-time. This change introduces significant challenges for both organizations and employees:

- **Space Utilization**: Companies need efficient systems to manage office spaces, avoiding underutilized or overcrowded areas.

- **Employee Experience**: Employees expect a productive and comfortable workspace when they come to the office, emphasizing the need for user-focused solutions.

- **Adaptability**: Organizations must implement systems that can evolve alongside changing work preferences and organizational structures.

Seat Booker addresses these challenges by offering a flexible and scalable solution. By enabling employees to reserve seats as needed, it fosters an adaptable workspace that balances personal and organizational needs. Additionally, it provides tools for optimizing space utilization, ensuring that offices remain both cost-effective and conducive to productivity. As Barrero et al. (2021) note, hybrid work models can lead to long-term savings and improved employee satisfaction when effectively managed, underscoring the value of platforms like Seat Booker. [3]

From a business and technical perspective, the Seat Booker project is designed to meet the growing demand for efficient workspace management in hybrid work environments. Its approach ensures a seamless integration of remote and in-office work, with features that address critical organizational needs:

- Optimizing Office Space: Companies can better manage their real estate, reducing operational costs tied to unused areas while preventing overcrowding.

- Enhancing Employee Productivity: By allowing employees to book workstations with specific configurations, Seat Booker supports tailored work environments that foster productivity.

· Improving Resource Management: Administrators gain access to tools for configuring layouts, monitoring availability, and ensuring necessary resources are in place, enhancing overall office management.

Through these features, Seat Booker not only adapts to the current demands of hybrid work but also prepares organizations for future shifts in workplace dynamics.

**Objectives Set for the Team:**
The development of the Seat Booker system was guided by clear objectives aimed at addressing the challenges of hybrid work environments while enhancing employee experience and organizational efficiency. Key goals included:

· **Developing a Flexible Workspace Reservation System:** Providing employees with a seamless way to reserve workstations for specific days, ensuring convenience and reducing conflicts in shared office spaces.

· **Creating Configurable Workstations:** Empowering administrators to define workstation features tailored to individual needs, such as additional monitors or adjustable desks, promoting productivity and comfort.

· **Designing Comprehensive Office Layouts:** Offering detailed floor plans with elements like printers, meeting rooms, and collaborative zones to help employees make informed choices when booking spaces.

· **Implementing Real-Time Booking and Availability:** Ensuring that the system accurately reflects live updates on workstation availability, allowing users to book or adjust reservations in real-time.

**Business and Technical Benefits:**
The Seat Booker project delivers value not only to end users but also to the organizations that adopt it. By leveraging modern technologies and user-centered design principles, it provides:

· **Increased Efficiency:** Streamlined management of office resources minimizes administrative overhead and improves the allocation of space and equipment, enabling organizations to optimize costs.

· **Enhanced Employee Satisfaction:** Flexibility in workspace reservations, coupled with access to well-equipped workstations, ensures employees feel supported in their work, boosting morale and productivity.

· **Scalable Infrastructure:** Utilizing tools like AWS, Terraform, and Docker ensures a scalable and resilient infrastructure that supports business growth. Terraform's infrastructure-as-code approach simplifies resource lifecycle management [4].

· **Robust Security and Authentication:** Advanced security measures, including Amazon Cognito and JWT, safeguard user data and ensure trust in the platform, meeting stringent privacy and security standards.

Overall, Seat Booker aligns with modern business practices, allowing companies to adapt to evolving work trends and maximize the potential of their office spaces.

## 1.2   Related Work

**Existing Solutions:**
In the context of workspace management systems, several existing solutions and technologies address similar challenges. These systems typically aim to optimize office space usage, enhance employee experience, and provide features such as booking, resource management, and real-time availability updates. Below is a brief overview of related works and how Seat Booker differentiates itself.

Products like Robin [5] and Envoy [6] offer workspace booking functionalities, focusing on hybrid work management. They provide seat reservation, layout customization, and analytics for office utilization.

· Strengths: Mature platforms with robust analytics and integrations.

· Weaknesses: These platforms often come with high costs, limited customization for smaller companies, and may lack adaptability for rapidly changing needs.

While existing solutions provide general workspace management, Seat Booker distinguishes itself through the following features and benefits:

- Customizable Workstation Elements: Unlike many existing platforms, admins can configure specific workstation components such as extra monitors or adjustable desks, tailoring workspaces to employee needs.

- Scalable, Cloud-Based Infrastructure: By leveraging AWS, Terraform, and Docker, the platform ensures seamless scalability, catering to organizations of varying sizes and growth stages.

**Design Considerations:**

The development of Seat Booker involved addressing specific challenges to deliver a robust, user-friendly system:

1. Technology Selection:

   - Frontend: React.js [7] for responsive UI and a seamless user experience.
   - Backend: Node.js [8] and Express.js [9] for handling real-time booking operations.
   - Database: AWS RDS [10] with MySQL [11] to ensure reliable data storage and retrieval.
   - Infrastructure: Terraform [12] and Docker [13] for scalability and efficient resource management.

2. Time Constraints: The team faced a limited timeline to deliver a fully functional MVP (Minimum Viable Product) while ensuring high-quality features. Developing a Minimum Viable Product (MVP) is a strategic approach that enables teams to validate product hypotheses with minimal resources, ensuring that the market desires the product before significant investments are made. This method accelerates learning and reduces wasted engineering hours [14]. By focusing on core functionalities, an MVP allows for the collection of maximum validated learning about customers with the least effort, facilitating informed decisions for future development.

3. Resources and Scalability: Balancing development with scalability, ensuring the system can accommodate future growth without overcomplicating the initial architecture.

4. User-Centric Design: Emphasis on intuitive navigation, with clear workflows for admins and employees, ensuring ease of use across diverse user groups.

By addressing these considerations and leveraging insights from existing solutions, Seat Booker provides a modern, scalable, and cost-efficient approach to workspace management, catering to the evolving needs of hybrid work environments.

## 1.3   Results

The Seat Booker project achieved significant technical and business milestones, delivering a functional and scalable system that addresses the challenges of hybrid work environments. Below is a detailed description of the results and functionalities implemented in the project:

**Functionalities Realized:**

The platform provides features tailored to both admins and users, ensuring seamless management of office spaces and reservations:

1. General Features:

   - User Authentication: Secure login functionality using Amazon Cognito and JWT, ensuring that only authorized users can access the platform.

2. Admin Features:

   - User Management:
     - Adding, editing, and deleting users.
     - Assigning specific spaces to users based on their roles or requirements.
   - Building Management:
     - Adding, editing, and deleting buildings with details such as addresses and cities.
   - Space Management:
     - Adding, editing, and deleting spaces within buildings.
     - Configuring space descriptions, including whether spaces have specific equipment such as monitors or liftable desks.

- Adding specialized areas within spaces, such as printer stations, to assist users in making informed reservations.
- Reservation Management:
  - Viewing and managing user reservations to ensure optimal space utilization.
  - Overriding or updating bookings when necessary for organizational needs.

3. User Features:

- Reservation Management:
  - Viewing personal reservations in a dedicated dashboard.
  - Booking specific spaces for desired dates, with real-time updates on availability.
  - Canceling reservations when no longer needed, with checks to ensure only the reserving user can cancel their booking.

**Business and Technical Objectives Met**

1. Business Objectives:

- Efficient Space Utilization: The platform enables companies to maximize the use of their office space by providing tools for admins to monitor and manage reservations effectively.
- Enhanced Employee Experience: Employees can reserve workspaces tailored to their needs, improving productivity and satisfaction.
- Flexibility for Hybrid Work Models: The system supports dynamic workplace arrangements, allowing companies to adapt to evolving work trends.

2. Technical Objectives:

- Scalable Architecture: The use of AWS and Docker ensures that the platform can scale with business growth, accommodating an increasing number of users and reservations.
- Real-Time Functionality: Real-time updates for booking and availability provide users with an accurate and seamless experience.
- Customization: Admins can configure spaces with detailed attributes (e.g., equipment and descriptions), offering a level of personalization not commonly available in similar solutions.
- Secure Authentication: Robust security measures, including JWT and Amazon Cognito, ensure user data privacy and system integrity.

# 2   CONCLUSION

## 2.1   Conclusions

The Seat Booker project successfully addressed the challenges of hybrid work environments by providing a flexible and user-friendly platform for office space reservation and management. The system's critical success lies in its ability to cater to both employee and administrative needs, delivering features such as customizable workstations, real-time booking updates, and efficient reservation management. These functionalities not only optimize office space utilization but also enhance employee satisfaction and productivity.

The project's technical achievements, including scalable infrastructure, secure authentication, and customizable configurations, demonstrate its readiness for deployment in modern workplaces. By offering a robust and adaptable solution, Seat Booker positions itself as a valuable tool for companies navigating the shift to hybrid work models.

## 2.2   Future Directions

As workplaces continue to adapt to hybrid models, our Seat Booker project has the potential to evolve further by introducing new features that enhance its usability and expand its functionality. These additions would improve user experience, foster collaboration, and support organizations in optimizing their workspace management:

- Add Buddies Feature: Introduce a functionality where users can add colleagues as "buddies" to see where they are seated. This feature would facilitate better coordination among coworkers, making it easier for teams or friends to choose seats close to each other.

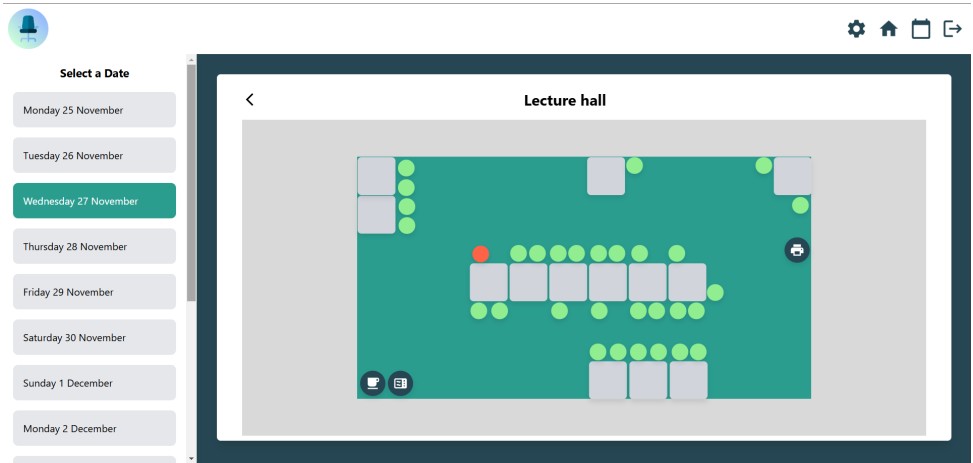

Figure 1: Booking page

- Automatic Check-In via Wi-Fi: Implement an auto-check-in feature that activates when users connect to the office Wi-Fi network. This addition would simplify the check-in process and help companies monitor attendance seamlessly.

- Parking Spot Booking: Expand the current reservation system to include parking spot bookings. This enhancement would allow employees to secure parking spaces in advance, reducing daily commuting stress and enhancing convenience.

- QR Code Scanning for Check-In and Reservations: Integrate QR code scanning functionality so users can scan a code to check in or reserve seats. This would streamline the process, especially for spontaneous bookings or when verifying reservations in real time.

- Team Reservations: Introduce the ability for teams to make collective seat reservations for collaborative work sessions. This feature would support teams in securing adjacent workstations for better collaboration and productivity.

- Enhanced Analytics and Reporting: Add features that provide companies with detailed insights into workspace utilization, employee preferences, and booking trends. This data could help optimize space planning and improve resource allocation.

- Mobile App Integration: Create a mobile application version of Seat Booker for easy, on-the-go seat booking and check-in functionalities.

- Multi-Language Support: Expand language options to make the platform more accessible to international users and companies with global offices.

- Integration with Calendar Services: Allow reservations to be automatically synced with popular calendar services such as Google Calendar and Microsoft Outlook for better scheduling integration

Implementing these features and enhancements would position Seat Booker as a comprehensive, user-friendly tool that adapts to modern workplace needs, promoting collaboration, flexibility, and overall productivity.

## 2.3   Acknowledgments

We would like to extend our heartfelt gratitude to mgr inż. Oleksandr Yeroshkin (MSc), our supervisor, for his invaluable guidance and support throughout this project. His expertise and encouragement were instrumental in helping us overcome challenges and achieve our objectives.

We also wish to thank Wroclaw University of Science and Technology for providing us with the resources, knowledge, and platform to develop this project. The university's emphasis on innovation and practical application of technology has been a source of inspiration and motivation for our team. Thank you for fostering an environment where creativity and collaboration thrive.

Finally, we express our deepest gratitude to our families and friends for their unwavering support, insightful suggestions, and active participation in testing and providing feedback during the development of this project. Your contributions have been invaluable in shaping the final outcome.

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
