# OpenReview forum: "Seat Booker Office Space Reservation System"
_pwr.edu.pl/Wrocław_University_of_Science_and_Technology/2024/ZPI_Day — Wrocław University of Science and Technology 2024 ZPI Day Submission_

### Official Review · Reviewer_xAMA · 2024-12-06
**Seat Booker - a review**

**Confidence:** 5
**Significance Of Results:** 5
**Overall Quality:** 5

**Compliance With Template:**

5: Very High Quality – The article contains all the required sections, which are written in a very detailed, clear, and error-free manner. The structure is professional and meets expectations, and the content adheres to the highest substantive and formal standards.

**Description Of Results:**

5: Very High Quality – The results are described in detail, clearly and comprehensively, supported by thorough evaluation, analysis, and convincing usage examples. The description meets the highest substantive standards.

**Feedback On Consistency:**

The abstract is consistent and done with care. The flow of the document is logical and seamless. On the other hand, the introductory section seems to be a kind of reptitive. To some extent it has been enforced by the structure of the abstract template, but repetiton of basically the same information in both "significant challenges for both organizations and employees", and "features that address critical organizational needs" should've been improved by, e.g. rephrasing or reference to the previous part of the text.

Moreover, the "Existing solutions" section should emphasise the strength of the product much stronger. Example of one of the weaknesses is the fact that authors state, that Robin and Envoy provide layout customisation, and, at the same time, for their tool: "Unlike many existing platforms, admins can configure specific workstation components", which could be interpreted as layout configuration. The differences should be more emphasised, but in the end the overall message is clear.

Other than that, the rest of the document is informative and meets the requirements. The references are not consistent regarding the style, which is the result of both scientific and more technical sources given there. Authors may want to either create the bibliographical entries for the technical sources, such as documentations, or, at least, group the sources in the bibliography starting with the books/papers, and then with the bare links.

**Potential For Development:**

The potential for development described in the abstract seems to be natural and is justified by expected user needs.

**Project Nature Evaluation:**

The project definitely is an engineering work, as it does solve a typical problem, by available means. Tools seem to be adequate to the task.

**Technical Language Precision:**

5: Very High Quality – The language is entirely appropriate for a technical report. All terms are used correctly and precisely, and the style is professional, clear, and coherent, without any errors or ambiguities.

---

### Official Review · Reviewer_9NjP · 2024-12-06
**The project presents a well-structured and functional system designed to address the challenges of hybrid work environments, with a clear focus on enhancing office space management and user experience. It effectively integrates technical solutions such as secure user authentication, space management, and reservation systems, while also highlighting potential for further development.**

**Confidence:** 5
**Significance Of Results:** 5
**Overall Quality:** 5

**Compliance With Template:**

5: Very High Quality – The article contains all the required sections, which are written in a very detailed, clear, and error-free manner. The structure is professional and meets expectations, and the content adheres to the highest substantive and formal standards.

**Description Of Results:**

4: High Quality – The results are described in detail and supported by usage examples or evaluations. The description is reliable but may lack full depth of analysis.

**Feedback On Consistency:**

Yes, the project description is consistent and logical. The problem analysis, presentation of results, and conclusions are well-aligned, providing a clear and coherent understanding of the project’s objectives, implemented features, and potential for future development.

**Potential For Development:**

Yes, the article clearly outlines several possibilities for further work and practical applications of the Seat Booker project. The proposed features, such as adding a “Buddies” feature, automatic check-in via Wi-Fi, parking spot booking, and enhanced analytics, demonstrate a clear vision for expanding the tool’s functionality to better serve modern workplace needs. These additions would improve user experience, enhance collaboration, and optimize workspace management, positioning the project for further growth and real-world application.

**Project Nature Evaluation:**

Yes, the Seat Booker project exhibits clear characteristics of an engineering work. It demonstrates the application of technical methods and technological solutions in creating a scalable and functional platform designed to optimize workspace management in hybrid work environments. The project incorporates features like secure user authentication, reservation management, and space management. These technical implementations showcase the project’s utility in solving real-world problems, particularly for organizations aiming to streamline space utilization and enhance the hybrid work experience.

**Technical Language Precision:**

5: Very High Quality – The language is entirely appropriate for a technical report. All terms are used correctly and precisely, and the style is professional, clear, and coherent, without any errors or ambiguities.

---

### Official Review · Reviewer_byaM · 2024-12-07
**Seat Booker Office Space Reservation System**

**Confidence:** 4
**Significance Of Results:** 3
**Overall Quality:** 3

**Compliance With Template:**

3: Average Quality – The article includes most of the required sections, but some may be incomplete, written in a general or unclear manner. The content is correct but requires further refinement.

**Description Of Results:**

3: Average Quality – The results are described with moderate detail. Some examples or evaluation elements are present but insufficiently developed or incomplete.

**Feedback On Consistency:**

- The problem analysis is clear and well-structured, but the link between the identified problem and the chosen methodology could be strengthened to ensure coherence.
- The analysis explains the problem's importance but could benefit from more specific examples to illustrate its relevance.

**Potential For Development:**

The project exhibits strong potential for development but would benefit from a more detailed exploration of practical applications and a roadmap for future work.

**Project Nature Evaluation:**

The project aligns well with engineering principles but could be improved by including a prototype or practical validation of the proposed solutions.

**Technical Language Precision:**

3: Average Quality – The language is mostly appropriate but may contain minor terminological or stylistic errors. Some statements might lack precision or require improvement for better readability.

---

### Decision · Program_Chairs · 2024-12-10

Accept (Poster)